# Natural Killer (NK) Cell Expression of CD2 as a Predictor of Serial Antibody-Dependent Cell-Mediated Cytotoxicity (ADCC)

**DOI:** 10.3390/antib9040054

**Published:** 2020-10-16

**Authors:** Jennifer J.-J. Tang, Alexander P. Sung, Michael J. Guglielmo, Lydia Navarrete-Galvan, Doug Redelman, Julie Smith-Gagen, Dorothy Hudig

**Affiliations:** 1Department of Microbiology and Immunology, Reno School of Medicine, University of Nevada, 1664 N. Virginia St., Reno, NV 89557, USA; Jia.jie.tang.12@gmail.com (J.J.-J.T.); asung@unr.edu (A.P.S.); mguglielmo@unr.edu (M.J.G.); lydia_ltng26@nevada.unr.edu (L.N.-G.); 2Department of Physiology and Cell Biology, Reno School of Medicine, University of Nevada, 1664 N. Virginia St., Reno, NV 89557, USA; dhudig@medicine.nevada.edu; 3School of Community Health Sciences, University of Nevada, Reno, 1664 N. Virginia St., Reno, NV 89557, USA; jsmithgagen@unr.edu

**Keywords:** ADCC, antibody-dependent cell-mediated cytotoxicity, CD2, CD16A, NK, natural killer cell, serial killing

## Abstract

NK cell ADCC supports monoclonal antibody anti-tumor therapies. We investigated serial ADCC and whether it could be predicted by NK phenotypes, including expression of CD16A, CD2 and perforin. CD16A, the NK receptor for antibodies, has AA158 valine or phenylalanine variants with different affinities for IgG. CD2, a costimulatory protein, associates with CD16A and can augment CD16A-signaling. Pore-forming perforin is essential for rapid NK-mediated killing. NK cells were monitored for their ADCC serial killing frequency (KF). KF is the average number of target cells killed per cell by a cytotoxic cell population. KF comparisons were made at 1:4 CD16pos NK effector:target ratios. ADCC was toward Daudi cells labeled with ^51^Cr and obinutuzumab anti-CD20 antibody. CD16A genotypes were determined by DNA sequencing. CD2, CD16A, and perforin expression was monitored by flow cytometry. Serial killing KFs varied two-fold among 24 donors and were independent of CD16A genotypes and perforin levels. However, high percentages of CD2pos of the CD16Apos NK cells and high levels of CD16A were associated with high KFs. ROC analysis indicated that the %CD2pos of CD16Apos NK cells can predict KFs. In conclusion, the extent of serial ADCC varies significantly among donors and appears predictable by the CD2posCD16Apos NK phenotype.

## 1. Introduction

Antibody-dependent cell-mediated cytotoxicity (ADCC) is critical for many monoclonal antibody (mAb) therapies directed toward tumors. ADCC involves a virally infected or tumor “target” cell with antigens, IgG1 or IgG3 antibodies that bind to the antigens, and effector cytotoxic cells with receptors for the antibody, reviewed [1]. Both natural killer (NK) lymphocytes and macrophages can be ADCC effector cells. NK cells mediate very fast cytotoxic destruction and over 75% of blood NK cells, including most subsets, have receptors for antibodies [2]. Despite the potency of NK ADCC, some patients’ tumors respond to mAb anti-tumor therapies while many do not [3,4,5,6]. Tumors can escape from attack by outgrowth of variants lacking the critical antigen [7] but there has been little attention given as to how the extent of variations in ADCC capacity could contribute to patient outcomes. Cytotoxic capacity differs from recognition of antibodies. Cytotoxic capacity, affected by amounts of cytotoxic proteins, can be limiting despite optimal recognition of “target” cells. Recognition can also be limiting and is affected by the properties of antibodies and by cellular Fc-receptor genetic variants that differ in affinity for the Fc of IgGs. These variants have been extensively studied for their effects on several anti-tumor immunotherapies, reviewed [8]. In this report, we focused on inter-individual variation in serial ADCC. Serial ADCC is the sequential killing of multiple antibody-coated targets by a single NK cell. It affects ADCC capacity. We tested for an immunophenotype that could predict high serial ADCC.

Serial NK cell-mediated killing can be investigated by time-lapse cinematography [9,10,11,12], by microscopy of single cells in microchip wells [10,13,14,15], and by a killing frequency (KF) approach [16]. KF is the average number of “target” cells killed per potential effector cell in a large population of effector cells. When the KF exceeds one there is serial killing [16]. Inter-donor variations in KFs were used to compare serial ADCC. Fc-receptor (CD16A) positive NK cells were used as the effector cell population. These receptor-positive cells had to be determined because not all NK cells have Fc-receptors. The effector cell population was left within the peripheral blood mononuclear cells (PBMCs) and used without further isolation. Previously, isolated NK cells showed comparable ADCC activity as unfractionated PBMCs (reference [17] Figure 4, doi:10.1016/j.jim.2017.11.002). This figure illustrates that ADCC by non-NK cells is negligible within the experimental conditions used. In order to quantify the number of ADCC effector NK cells in the whole PBMCs, a panel of fluorescent antibodies toward specific cell markers and Trucount^R^ beads were used with flow cytometry. The population of NK ADCC effector cells was specifically identified as being CD3negCD16posCD7posCD33negCD45pos, which effectively excludes T cells, B cells, and monocytes.

The KF approach has one advantage over time-lapse cinematography and microchip microscopy which provide information about the maximal serial killing by individual cells. KFs analyze thousands of killer cells and include diverse NK subpopulations. There are over 6000 subgroups of NK cells, each with different combinations of receptors and the ratios of the subgroups vary from donor to donor [2]. Most of these NK subgroups have CD16A receptors for the Fc of antibodies. However, the subgroups differ in additional, non-Fc receptors that can regulate ADCC [18,19,20]. Variability in the frequencies of the subgroups with these receptors contributes to variability in cytotoxic capacity among donors [2]. In the present study, there were thousands of NK cells per well, six effector to target (E:T) ratios, and quadruplicate wells for each effector to target ratio (E:T),. When comparing human subjects, it is important to utilize a large enough sample of NK cells so that the complex nature of each individual’s NK cell population is truly represented.

In contrast to KF assays, time lapse cinematography and microchip assays track only a few cells or a few hundred cells, respectively. However, information from cinematography and microchip experiments can help in the interpretation of the KF ADCC assays. Time lapse cinematography indicated that, in NK modes without antibody, there is heterogeneity in serial killing. Some cells within the effector cell populations bound to targets without killing them while the cytotoxic cells killed 1, 2, 3, or even up to 5–7 times [16,21]. Microchip experiments also indicated cellular differences in killing when the NK cells are in ADCC mode. A substantial fraction of the potential killers with CD16A killed no targets at all while potent ADCC cells killed up to three target cells before they stopped killing [14]. KF assays average the activity of all the cells with receptors needed for recognition of the targets, including those cells that kill and those that do not kill. Assays with KF values >1.0 indicate that there is serial killing by a cell population but do so without revealing how many cells are actually killing or how many times each killer kills. In this study, the KF assays indicate the overall net serial ADCC capacity of the CD16Apos NK cells.

The inter-donor variability of KFs that we observed provided the data needed for prediction of serial ADCC by immunophenotypic characteristics. Three proteins, CD16A, CD2, and perforin, were selected for phenotyping. CD16A is the NK cell receptor that binds the Fc-region of IgG antibodies in ADCC. Its variability among donors is partially determined by genetics. The *FCGR3A* gene has two alleles that encode CD16A at AA158: one that encodes valine (V) and has twice the affinity for the Fc-IgG and two-fold more cell surface CD16A than the other, a phenylalanine (F) variant [8,22,23,24]. CD16A is lost by proteolytic cleavage during killing [25] and upon IL-2 activation [26] which makes CD16A a candidate receptor to cause variations in serial ADCC. Because of the higher affinity and cellular expression, we anticipated that cells with V alleles (V/F and V/V) would mediate more serial killing than F/F cells.

CD2 is a costimulatory molecule that generates signals to increase the cytotoxicity of NK cells [27,28,29,30], reviewed [31]. CD2 physically associates with CD16A [32]. Co-engagement of CD2 and CD16A will result in a Ca^2+^ influx and augment anti-CD16A redirected lysis by NK cells [18]. Among healthy adults, the % of NK cells that are CD2positive (%CD2pos) varies widely, e.g., from 16% to 90% (median 66% for 103 donors, D. Redelman, unpublished results from a study of healthy adult civilians that was funded by the US Office of Naval Research). Variability is needed as a basis for a predictive test.

Perforin is a critical pore-forming protein that is stored in intracellular cytotoxic granules of T and NK cells, reviewed [33] and released during killing. While only a few granules are necessary for a killing event [34] and there are many cytotoxic granules per NK cell, depletion of perforin does occur upon serial re-stimulation of NK cells [35]. Perforin levels in NK cells also vary among donors [36,37], making the three proteins, CD16A, CD2, and perforin, candidates to limit NK cell-mediated serial ADCC.

Here we report ADCC killing frequencies by unstimulated freshly isolated NK cells that can be as high as an average of four dead targets per killer cell. This observation indicates that substantial serial ADCC can be mediated by NK cells before they lose their Fc-receptors. The CD16A expressed per NK cell and the %CD2pos of the CD16A-positive NK cells varied widely among the 24 donors of this study, providing a range for correlations with serial ADCC. Excess targets favored increased serial killing and increased KFs. One effector to target ratio, 1:4, was used for inter-donor KF comparisons. Serial killing correlated best with the percentage of CD16Apos NK effector cells that expressed CD2. Receiver operating characteristic (ROC) analysis indicates that the %CD2pos of CD16Apos NK cells may be suitable as a test to predict serial ADCC. These observations indicate that CD2 immunophenotyping of NK cells may be worthy of consideration to select patients for antibody-directed anti-tumor therapies.

## 2. Materials and Methods

### 2.1. Human Subjects

The human subjects were the healthy family members and additional controls from a clinical study [17]. Citrated blood was drawn in Salt Lake City, UT, USA, and shipped overnight to Reno, NV, USA, where PBMCs were isolated [38]. Use of human subjects was approved by institutional review boards for the Bateman Horne Center and for the University of Nevada, Reno School of Medicine. Written informed consent was obtained from the blood donors. The ages, sex, and CD16A genotypes of the blood donors are presented in Table 1. All blood donors were Caucasian. Blood samples were coded in Salt Lake City. ADCC and EC50 assays, CD16A NK cell counts, immunophenotyping, and CD16A genotypes were determined with the coded samples and decoded after completion of the experiments.

### 2.2. Preparation of Peripheral Blood Mononuclear Cells (PBMCs)

The study involved 15 shipments with 2-6 blood samples per shipment. Forty milliliter of blood was divided as follows: 8 mL for DNA extraction (PAXgene^®^, Qiagen, Germantown, MD, USA, a BD company) and the remaining 32 mL into citrated tubes. The PBMCs were cultured overnight without stimulation [39] at 1–2 × 10^6^ cells/mL in 90% Dulbecco’ s complete media containing high (4.5 g/L) glucose and L-glutamine (Corning), 10% fetal calf serum (Atlanta Biologicals, Atlanta, GA, USA), 10 mM hepes (Sigma-Aldrich, St. Louis, MO, USA), and 1% penicillin-streptomycin (Sigma-Aldrich). Culture conditions and assay media were standardized with a single lot of fetal calf serum and one lot of tissue culture flasks (Biolite, Thermo Scientific, Waltham, MA, USA) throughout the study.

### 2.3. ADCC Assays

#### 2.3.1. ADCC Methods

The ^51^Cr-release ADCC assay [40] has three important features: (1) MHC class I-negative Daudi target cells (to avoid variations contributed by KIR engagement with MHC-I proteins); (2) a type 2 [41] anti-CD20 monoclonal antibody that is poorly cleared from the membranes of B cells (so that Daudi B cells could be pretreated with antibodies and washed to prevent competitive ADCC by B cells within the PBMCs that would occur if anti-CD20 antibody were present in the assays); and (3) use of unfractionated PBMCs with TruCount^®^ beads to determine the numbers of CD16A receptor-positive effector (E) NK cells within the PBMCs (to reduce NK cell losses that would occur during further isolation). The method is illustrated in a graphical summary [17] at https://ars.els-cdn.com/content/image/1-s2.0-S0022175917304295-fx1_lrg.jpg.

ADCC toward the Daudi cells was mediated by six different concentrations of CD16Apos NK cells. Daudi cells clear some of the type 2 anti-CD20 antibody during the 4 h assay; however, ADCC was unaltered when freshly labeled and 4 hr-preincubated at 37 °C targets were compared (Hudig et al., unpublished results). The Daudi lymphoma target cell line [42] from the ATCC (catalog # CCL-213) was routinely tested and negative for mycoplasma. Daudi cells were labeled with 0.5 mCi Na^51^CrO_4_ (Perkin Elmer, Waltham, MA, USA), pretreated with an antigen-saturating concentration (1 µg/mL) of Fc-engineered, non-fucosylated anti-CD20 monoclonal antibody obinutuzumab (Gazyva^®^) [43,44,45] for 0.5 h at room temperature and then washed 5 times to remove unbound antibody. PBMCs containing the NK effector cells were diluted two-fold in quadruplicate wells in 96-well V-bottom plates (Costar 3894) to create six CD16Apos NK effector to target cell (E:T) ratios.

Daudi cells (with or without mAb), for ADCC or for killing in NK mode in the absence of antibodies, respectively, were added at 10,000 cells in 0.1 mL to each well. Plates were centrifuged for 3 min at 1000 rpm and incubated for 4 h at 5% CO_2_ and 37 °C. After incubation, plates were centrifuged for 10 min at 1200 rpm and 0.1 mL of cell-free supernatant was counted for ^51^Cr-release in a Packard Cobre II gamma counter. The percent specific release (SR) was calculated using the formula
% SR = [(Experimental counts − Spontaneous Release)/(Max − Spontaneous Release)] × 100.

Spontaneous release is the leak rate of targets without PBMCs and the Max is the radioactivity released by targets lysed with 1% SDS. NK activity to Daudi cells without antibody was negligible (<~5% at the highest E:Ts).

Percent ADCC was plotted as linear cytotoxicity with y = % specific ^51^Cr release vs. x = the log_10_ of the six TruCount^®^ CD16Apos NK effector cell to Daudi target cell ratios. The linear cytotoxicity was used to calculate y = mx + b, with the lytic slope = m, x = log_10_ of the E:T and b = the y intercept. The *p* values for linearity were <0.05, with R^2^ values >0.8. Because the slopes of the cytotoxicity varied among donors, it was necessary to select one E:T (1:4) for inter-donor comparisons of killing frequencies.

#### 2.3.2. Killing Frequencies (KFs)

Killing frequencies are the number of “target” cells killed per cell within a cytotoxic cell population [16]. In this study, KFs were calculated as the number of Daudi cells killed per CD16Apos NK cell. The KFs can also be calculated by dividing the % of cells that are killed at a given E:T by the % of target cells that would be killed if each CD16Apos NK (at that E:T) killed one target. For example, at an E:T of 1:4, if every CD16Apos NK killed one target cell, one out of four available targets would be killed: (1 ÷ 4) = 25%. If the observed killing (at E:T = 1:4) was 50%, then the KF would be (50% ÷ 25%) = 2, indicating two targets were killed per “E” CD16Apos NK cell and that serial ADCC killing had occurred (see Graphical Abstract). To repeat, ADCC KF is calculated by dividing the number of antibody-coated Daudi target cells killed by the number of CD16Apos NK cells present. The KFs increased as the available target cells increased: the highest KFs were always at the lowest E:T for each donor (Appendix A
Figure A2).

Killing frequencies at one E:T, 1:4, with 2500 effectors and 10,000 targets, were used for the inter-donor comparisons. The numbers of CD16Apos NKs (Es) in the wells were based on their Trucounts^®^ within the PBMCs. As a result, the assays lacked exact 1:4 E:Ts in the wells but always had E:Ts that were below and above 1:4 and that were always in the range of linear cytotoxicity. The % killing at 1:4 was determined algebraically from the linear cytotoxicity described in method 2.3.a. It was calculated as y = mx + b, with y = % killing, m = slope, x = log10 of 0.25 [an E:T of 1:4], and b = the y intercept. Finally, in order to obtain the number of cells killed at an E:T of 1:4, because there were 10,000 Daudi cells per well the % killed was multiplied by 10,000. This number of killed cells was divided by 2500 (the number of CD16Apos NK cells present) to obtain the KFs at 1:4.

#### 2.3.3. EC50s for NK Recognition of Target-Bound Antibody

The EC50s (effective concentrations of antibody needed for 50% of maximal ADCC [46]) were determined with one concentration of PBMCs with four-fold dilutions of obinutuzumab in the assays at final concentrations from 0.04 to 625 ng/mL. EC50s were determined at 4 h, with duplicate or triplicate wells for each antibody concentration.

### 2.4. TruCount^®^ Determination of the Numbers of CD16A-Pos NK Cells in the ADCC Assays

Counts of the CD16A-pos NK cells within the PBMCS were as previously described [17]. Fifty-microliter aliquots of PBMCs were added to TruCount^®^ tubes (Becton Dickenson no. 340334 [47]) with fluorescent beads, labeled for 30 min with antibodies, fixed, and analyzed without washing to count beads and cells. The cells were labeled with PacBlue anti-CD45 (clone HI30) to identify all cells; FITC-anti-CD3e (clone UCHT1) to identify T cells; FITC-anti-CD7 (clone CD7-6B7), APC-Cy7 anti-CD56 (clone HCD56), and PerCP-anti-CD16A (clone 3G8) to identify ADCC effector cells, and PE-Cy7 anti-CD33 (clone P67.6) to identify monocytes, purchased from BioLegend (San Diego, CA, USA). The mAbs were all mouse IgG1 that do not bind to human CD16A. The ADCC effector NK cells were distinguished as CD3negCD7posCD16posCD33negCD45pos and CD56variable. Inclusion of anti-CD7 was critical to distinguishing the subgroup of CD56negCD7posCD16Apos NK cells [48] from CD56negCD7negCD16Apos monocytes (that are largely CD33pos [49]). Cells were analyzed the same day as the ADCC assays using a BD Biosciences Special Order Research Product LSR II analytical flow cytometer with a high throughput sampler unit. The data were assessed with FlowJo software (FlowJo, LLC, Ashland, OR, USA).

### 2.5. Immunophenotyping and Staining for CD2 Counter-Ligands

To monitor the %CD2pos cells and CD2 MFIs, PBMCs were stained with PacBlue anti-CD45, FITC-anti-CD3e FITC T cells and FITC-anti-CD91 (clone A2MR-alpha2) to identify monocytes, BV650-anti-CD19 (clone HIB19.11) to identify B cells, AF647-anti-CD16A and PE-anti-CD2 (clone RPA-2.10), all purchased from BioLegend, San Diego, CA, USA, except for the anti-CD91 that was purchased from BD BioSciences, San Jose, CA, USA. To monitor intracellular perforin, cells were first stained for extra cellular proteins with PacBlue anti-CD45, PE-Cy7 anti-CD3e, and PE-Cy7 antiCD33, the same clones as above, then fixed and permeabilized using IntraPrep reagents (Beckman-Coulter, Indianapolis, IN, USA), and stained with FITC-anti-perforin (clone delta-G9) or a FITC-isotype control (clone MPC-11), from BioLegend. Cells were analyzed on the day of labeling for perforin expressed by the NK cells. The lots of labeled antibodies were kept constant throughout the study to reduce inter-experimental variability. Daudi cells and K562 cells (ATCC^®^ CCL-243™) were stained for counter ligands of CD2 with PE-anti-CD15 (clone SSEA-1) or PE-anti-CD58 (clone TS2/9), from BioLegend.

### 2.6. Genotyping of FCGR3A Alleles Encoding CD16A F and V Variants

CD16A genotypes at AA158 were determined by Stephen K. Anderson, Ph.D., by PCR and DNA sequence analysis at the Frederick National Laboratory for Cancer Research, Frederick, MD, USA and additionally by flow cytometry. Amplicons of the *FCGR3A* gene that excluded the *FCGR3B* gene were generated with forward and reverse PCR primers, (5′ to 3′) for CD16 (TCCTACTTCTGCAGGGGGCTTGT) and (CCAACTCAACTTCCCAGTGTGATTG), respectively. The amplicons were sequenced using Sanger methodology. The F/F genotype was also distinguished from V/F and V/V genotypes by flow cytometry [17] using the clone MEM-154 anti-CD16 mAb. MEM154 reacts with the CD16A 158 V but not the 158 F [50] and also reacts with CD16B (that has only the valine form). PBMCs were labeled with an antibody panel: FITC-anti-CD3e (cloneOKT3); PE-anti-CD16A or PE anti-CD16A 158V selective-(MEM154); BV605-anti-CD19; PacBlue anti-CD45; FITC-anti-CD91 and APC-Cy7-anti-CD56, clones and sources previously indicated except for MEM154 mAb (Pierce Chemical Co, Rockford, IL, USA).

### 2.7. Statistical Analyses

ADCC measurements and linear correlations were determined with the Excel Analysis Tool Pack, using best fit for linear regressions to determine KFs. Student’s t-tests in Excel were used to compare the groups of donors. Excel and GraphPad Prism 7 (San Diego, CA, USA) were used for illustrations. Predictive potentials of the phenotypes were assessed using receiver operating characteristics (ROC) analyses [51] with SAS software version 9.4. The comparison of %CD2pos and CD16A MFI ROCs was made by Mann-Whitney two-sample rank measure (a generalized U statistic).

## 3. Results

### 3.1. Killing Frequencies Indicate Serial ADCC and Inter-Donor Variability

A few technical aspects [17] were critical to this study. The “effector” (E) cytotoxic lymphocytes were freshly isolated, unstimulated blood NK cells that express CD16A receptors for antibodies. The “target” (T) lymphocytes were Daudi B lymphoma cells bound with antibodies. It is important to realize why Daudis were chosen and how they were prepared. Daudis lack MHC class I proteins (which could increase vulnerability to NK but Daudis are nonetheless poor targets for NK activity without antibodies). Without MHC-I, ADCC to Daudis will be unaffected by KIR-MHC-I inhibition. The antibody, obinutuzumab anti-CD20, was produced without fucosylation (to better support ADCC [52]) and is Fc-engineered for high affinity for CD16A. Obinutuzumab, as a type-2 anti-CD20 mAb, will remain on the surface of the lymphoma cells rather than be endocytosed like a type 1 anti-CD20 mAb. ^51^Cr-radioactive Daudi cells were obinutuzumab-labeled and washed free of excess antibody so that unfractionated PBMCs could be used as the source of the NK effectors. If obinutuzumab were in the assay, the non-radioactive B cells within the PBMCs would compete and reduce killing of the ^51^Cr-Daudis. Use of unfractionated PBMCs permitted maximal NK cell recovery with minimal handling of the effector cells.

Killing frequency (KF) is a term [16] used for the average number of cells that a known number of cells in a cytotoxic lymphocyte population will kill in a fixed period of time. Killing frequencies greater than 1.0 represent serial killing. Lower KFs could indicate that no serial killing takes place or they could indicate that serial killing occurs but by only a fraction of the cells within the cytotoxic cell population that are engaged in killing. KFs are useful for inter-donor comparisons because multiple donors can be assessed concurrently, thereby reducing inter-experimental variability. Only a fraction of the CD16Apos NK cells actually killed when observed by microscopy [14,16] so these KF numbers represent an underestimation of serial killer activity.

The KFs can be directly determined for each E:T in the ADCC assays, by dividing the % target cells killed by the % dead cells that would be expected under the assumption that each cell in the cytotoxic cell population kills only one target cell. At an E:T of 1:4 one round of killing would kill 25% of the targets which would be a KF of 1.0. When the killing at this E:T was 48%, the KF was 1.9 (illustrated in Figure 1A). We selected a 4 h time point for determination of the KFs as ADCC was nearly complete at this time (see Appendix A
Figure A1).

### 3.2. ADCC Killing Frequencies of 24 Donors

KFs increased as the number of available target cells increased per CD16Apos NK cell, as illustrated in Appendix A
Figure A2. Six donors, including the one illustrated, had serial killing that averaged 4 or more dead targets per CD16Apos NK cell at their highest E:T ratios (of >1:28). At the highest target ratios available for each donor, with average E:Ts of 1:18, 21 of 24 donors (88%) had KFs >1.5, providing evidence for widespread serial ADCC. The pattern of increased serial killing with increased availability of targets was reported previously for NK killing with or without antibodies [16].

The KFs at a 1:4 ratio of CD16Apos NK to Daudi cells were selected for inter-donor comparisons. The 1:4 ratio was selected because it was within the six E:T ratios of the ^51^Cr-assays for all the donors (as indicated by the red arrow, Figure 1A). At the E:T of 1:4, the KFs ranged between 1.1 and 2.2. The mean KF was 1.6 +/− 0.3 for all donors, with mean KFs of 1.6 +/− 0.4 for females and 1.7 +/− 0.3 for males. There was little evidence for correlation of KFs with age (Figure 1B) or gender (Figure 1C).

### 3.3. The Effect of CD16A Genotypes on ADCC KFs

We compared F/F vs. V/F and V/V genotypes. The V/V and V/F genotypes were combined for comparisons because the effect of the V allele dominates in heterozygotes. The mean KF for the F/F cells was 1.5 +/− 0.4, while the mean KF for the V/F and V/V donors was 1.7 +/− 0.3 (Figure 2A), indicating similar serial ADCC. The CD16A genotypes did have effects that have been reported before, indicating that our samples have typical properties. The F/F genotype required more antibody to support lysis, as indicated by the EC50s (effective concentration of antibody to support 50% lysis (Figure 2B). The lower CD16A MFIs of the F/F genotype indicated lower numbers of CD16A receptors per NK cell (Figure 2C), consistent with an earlier report using the 3G8 anti-CD16A antibody [22]. The lack of CD16A genotypic effects on serial ADCC was determined with optimal anti-CD20 antibody. Obinutuzumab, a type 2 anti-CD20 antibody, was non-fucosylated, Fc-engineered to improve binding to CD16A, and at saturating antibody concentrations on the Daudi cells. Thus, inter-donor differences in KFs occurred under antibody conditions that over-rode CD16A genotypic effects (e.g., EC50s) that can occur at low antibody concentrations.

### 3.4. Assessment of NK Phenotypic Markers to Predict KF

To predict serial killing, we divided the samples into two groups, one with high and one with low KFs. We used a KF of equal to or greater than 1.5 as the dividing cutoff. There were 14 donors in the high KF and 10 in the low KF groups, with means of 1.8 +/− 0.02 and 1.3 +/− 0.04, respectively (Figure 3A). CD16A expression was similar for both groups (Figure 3B). There were no detectable differences in perforin levels (Figure 3C). Also, *within* each group, expression of CD16A and perforin failed to correlate with KF (not illustrated). In marked contrast, the percentage of the CD2pos of CD16Apos NKs was much greater for the donors with high KFs (mean 76.6%, *p* < 0.001) than with low KFs (mean 53.3%) (Figure 3D). The amounts of CD2 expressed by these CD2pos cells were similar for both groups (Figure 3E) and indicate that the CD2posCD16Apos NK cells have sufficient CD2 to augment signaling, regardless of their KF status.

### 3.5. Tests of CD2 Immunophenotype to Support Prediction of Serial ADCC Capacity

The percentage CD2pos of CD16Apos NK cells correlated positively with KFs (Figure 4A). The KF-%CD2pos correlations applied to both CD16A F/F and V/F and V/V donors. The statistical significance of the V donors was affected by one outlier, but the significance for all donors was *p* < 0.001 (black dotted line).

ROC analysis indicted good predictive value (Figure 4B) for KFs by the %CD2pos of CD16Apos NK cells. ROC tests evaluate the reliability of a test, using a series of cutoffs based on experimental data for trial evaluation. Donors were divided into two “true” groups (see Figure 3A) with high vs. low KFs. The %CD2pos values used to determine sensitivity and specificity ranged between 51% and 96% for the 24 donors. The symbols on the graph illustrate these sensitivity and specificity values. The area under the curve (AUC) represents the predictive value and was 0.89 (*p* < 0.001) (Figure 4B). An AUC of 1 is a perfect test and an AUC of 0.5 is without any predictive value; 0.89 indicates good predictability. The inflection point for 100% sensitivity occurred at 60% CD2pos of CD16Apos NK cells and indicates a suitable cutoff to predict high ADCC KFs.

Quadrant analysis of a test using the 60% CD2pos cutoff provides additional information. Only two false tests occurred for the 24 donors (Figure 4C). The predictive values were excellent: the positive predictive value was 0.88 and the negative predictive value was 1.0. Suppose that this CD2 test was applied to assign only donors with high KFs for immunotherapy. All the true high KF patients would be identified and receive therapy. Two false positives with high %CD2pos cells (but low KFs) would be misidentified and undergo a therapy that might offer them less benefit.

In contrast to the value of the %CD2posof CD16Apos NK cells to predict KFs, the CD16A MFIs of the effector cells were less suitable (Figure 4B). The AUC of 0.71 (*p* < 0.01) indicates potential predictive value but was lower than the AUC of 0.89 for %CD2pos. Quadrant analysis with the best cutoff for CD16A MFIs indicated five false predictions for the 24 donors (not illustrated). The *p* value for differences between the %CD2pos and CD16A MFI ROCs was 0.08. The data indicate that %CD2pos of CD16Apos NK cells may be a suitable predictive test for serial ADCC capacity and is likely to be better than CD16A MFIs.

### 3.6. Potential for CD2—Counter Ligand Engagement During ADCC

CD2 engages ligands and then transduces signals that support T-cell proliferation [53,54,55] and NK responses [56]. For NK cells, CD2 engagement is important for antibody-induced cytokine and cytotoxic responses by “adaptive” NK cells and is less important for “conventional” NK cells [57,58]. CD2 physically associates with CD16A [32]. Consequently, CD2 may be drawn into an NK-target cell synapse that is initiated by CD16A binding to antibodies on target cells even without CD2-ligand engagement. Human CD2 has two ligands, CD15 and CD58. We detected no CD15 and only low levels of CD58 on Daudi “target” cells (Appendix A
Figure A5A). Anti-CD58 antibodies failed to affect ADCC (Figure A5C) even though they reduced NK activity toward K562 cells which have substantial CD58 (Figure A5B). Thus, there was little evidence for important CD2-CD58 or CD2-CD15 interactions during the ADCC assays toward Daudis.

## 4. Discussion

To the best of our knowledge, we are the first to characterize human variability in serial ADCC capacity. We report variability in serial ADCC by an unstimulated NK cell population that represents the state of NK cells in vivo. We investigated the capacity of the entire NK cell population and expanded information provided by others who studied serial ADCC by cinematography of individual NK cells [14,16]. Our observation of serial ADCC was facilitated by an experimental design that resulted in the absence of spontaneous NK to the Daudi targets, by lack of spatial constraints (because the target cells were non-adherent and could form a multi-layer cell pellet), and by a highly engineered mAb. Serial killing with a KF = / >1.5, at high target excesses was unequivocal for 88% of the donors. At an absolute minimum of serial killing, 50% of the CD16Apos NK cells would have had to kill twice. KFs of 4 or higher were observed for 25% of the donors with these high target cell excesses. This activity is notable since it is widely believed that ADCC serial killers are rare due to the NK loss of CD16A Fc-receptors during ADCC [25,59]. We observed that ADCC was of short duration, terminating after about 4 h, consistent with limited rounds of ADCC before the loss of Fc-receptors.

Efforts to understand the inter-donor variation in serial ADCC led to three new insights. [1] The serial ADCC appeared to be independent of CD16A AA158 genotypes (Figure 2A), which was contrary to our initial expectation. Our expectation was that both F/V and V/V genotypes would have higher KFs than the F/F genotype because a) fewer of the high affinity V Fc-receptors would be needed and b) the greater density of the V Fc-receptors would leave more receptors after the first kill available to kill again. Information is still needed as to whether these insights apply to mAbs that contain a mixture of fucosylated and non-fucosylated antibodies and/or that retain unmodified native Fc’s. [2] Perforin levels were similar for the high vs. low KFs, indicating that sufficient perforin was available for the NK cells with low KFs. [3] The most notable finding was that the %CD2pos NK cells had predictive value for KFs.

Few previous investigators have addressed serial ADCC. Perhaps researchers felt that serial ADCC was unlikely because the CD16A receptor is cleaved away by metalloproteases during killing. Two other groups have observed serial ADCC. Romain et al. [14], using microscopy of single NK cells in microchip wells, observed serial ADCC of up to three target cells per effector cell. This team, using mouse EL4 targets transfected to express human CD33 as a ligand and an Fc-engineered anti-CD33 mAb, found that 28% of the NK cells killed two or three targets. Drs. Rauf Bhat and Carsten Watzl [16] used ^51^Cr-release assays to observe antibody-dependent increases in killing of 722.221 B lymphoma cells. The ADCC was supported by rituximab mAb that has a native Fc. There was also some spontaneous NK killing of the target cells. For three donors, they observed an average NK KF (without antibody) of 1 and an average ADCC KF of 3 (that included the NK activity) at 4 h. These findings [16] are comparable to the results reported here. The two studies cited [14,16], together with this report, indicate that substantial serial ADCC occurs toward different tumor target cells and with antibodies that have either native or engineered Fc domains.

We wanted a simple test for inter-donor variability that could be used for clinical applications. There is a real need for a simple predictive test to aid in selection of patients who are most likely to respond to mAb anti-tumor therapies. A survey of recent papers on the subject reveals that only 25–30% of HER-2/neu—positive breast cancer patients responded to trastuzumab [3,6], only ~54% of non-Hodgkin lymphoma patients responded to rituximab [4], and only 28% of metastatic colon cancer patients responded to cetuximab [60]. For trastuzumab and cetuximab, considerations for patient selection are complicated because these antibodies block receptors for growth factors as well as support ADCC. Each tumor may differ in terms of growth arrest vs. death by ADCC. Even in the face of these obstacles, the variability of in vitro serial ADCC correlated so well with the %CD2pos of CD16Apos NK cells that this phenotype could be considered as a potential test to help identify the patients who may respond best in anti-tumor immunotherapies directed by mAbs that bind to the tumor cells. The ROC AUC value of 0.98 is encouraging. It should be noted that differences attributable to a single variable are easier to observe in a homogeneous population than in a heterogeneous one. The population of this study was entirely Caucasian and from Utah, USA, where most of the population is of non-Finnish northern European descent [61]. It will take assessment of serial ADCC from geographically and socio-economically diverse donors to determine if the predictive value of the %CD2pos of CD16Apos NK cells applies to a more diverse population. Selection for serial ADCC efficacy is of great importance, since CD16A-engagement can reduce subsequent general NK activity after anti-CD20 immunotherapy [62].

The role of CD2 in serial ADCC is unclear; however, properties of CD2 underscore several potential roles. CD2 is involved in NK cell “priming” by NK CD2-monocyte CD15 interactions that increase subsequent cytotoxic activity [63]. The overnight culture of PBMCs in this study provided an opportunity for NK priming by CD15pos monocytes. It would be worthwhile to isolate NK cells prior to overnight culture to assess potential priming. CD2 increases signal transduction that directs cytotoxic degranulation [18,32]. However, CD2 engagement alone is insufficient to trigger lysis [18]. Grier [32] found that CD2 and CD16A are non-covalently associated via extracellular domains and that, curiously, CD2 can cause CD16A to signal even where there are no antibodies available to engage CD16A. They demonstrated that NK CD2—K562 CD58 interaction coincidentally recruited CD16A into the immunological synapse without involvement of antibody. In the present study, the Daudi cells had low CD58 (and undetectable CD15). We were unable to block ADCC with anti-CD58 antibodies, indicating that CD2-ligand engagement was probably unnecessary during killing. Finally, there is evidence that CD2 participates in ADCC by “adaptive” NK cells [57,58]. To date, information is lacking as to whether adaptive NK cells are serial killers.

## 5. Conclusions

In summary, we conclude that there are substantial differences in serial ADCC among human donors. These differences appear to be predictable by tests for %CD2pos of CD16Apos NK cells. Prediction of serial ADCC may be of clinical value to understand variations in patient responses to anti-tumor monoclonal antibodies.

## Figures and Tables

**Figure 1 antibodies-09-00054-f001:**
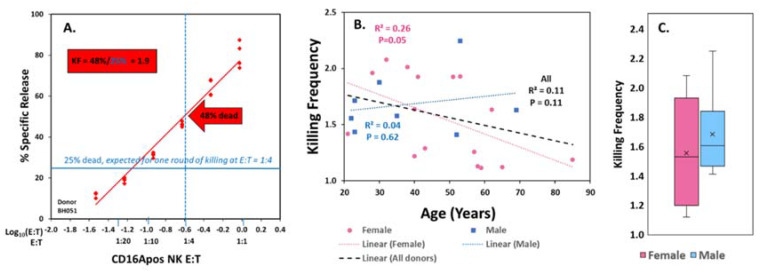
**ADCC killing frequencies (KFs).** (**A**) **Methodology.** KFs for comparisons were determined at the CD16Apositive NK cell (E) to antibody-labeled target (T) cell ratio (E:T) of 1:4. The KF of 1.9 is indicated in the red box, the 48% target cells killed (arrow) divided by 25% (indicated by the blue horizontal line for death predicted for a single round of killing by every receptor-bearing cell). (**B**) **and C. Individual variation in KFs.** Variations in KFs at E:T 1:4 for 24 donors. (**B**) **KFs vs. age.** The *p* values are indicated for linear fit. The *p* for women was influenced by inclusion of a very elderly subject. (**C**) **KFs vs. gender.** The boxes illustrate the 2nd and 3rd quartiles, the X’s indicate the mean values, the bars indicate the medians, and whiskers indicate the values for lower 1st and upper 4th quartiles. Points would indicate outliers. Other whisker plots in this report follow this format.

**Figure 2 antibodies-09-00054-f002:**
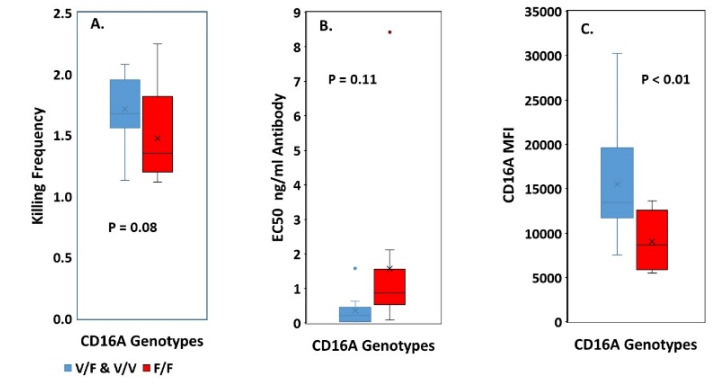
**Effect of CD16A genotypes on serial ADCC.** The V/V and V/F genotypes were combined for comparisons because the effect of the V allele dominates in heterozygotes. There was only one V/V, 11 V/F, and 12 F/F donors. *p* values represent 2-way T tests. (**A**) **KFs at 1:4 E:Ts.** The means (X’s) and medians (bars in the boxes) were similar for the genotypes. (**B**,**C**) **Effects of CD16A genotypes on NK function and phenotype.** (**B**) **EC50s.** More antibody was required to support ADCC by the F/F donors, which was nearly significant *p* = 0.055 by a one-way T test, and is consistent with previously reported EC50s for the F/F genotype [23,46]. **Blue and red dots** indicate outlier values. Representative EC50 determinations of the genotypes are illustrated in Appendix A
Figure A3. (**C**) **CD16A expression.** The V/F & V/V donors had ~1.7-fold more CD16A per cell than the F/F donors.

**Figure 3 antibodies-09-00054-f003:**
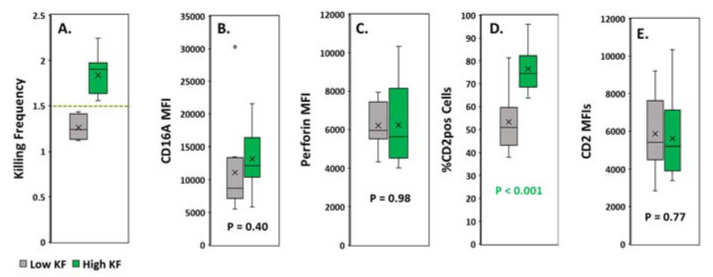
**Assessment of CD16A, perforin and CD2 expression** in CD16A-positive NK cells with high vs. low killing frequencies (KFs). (**A**) **Donors were divided into two groups by KFs** using a cutoff value of 1.5 (marked by the dashed line). (**B**–**E**) **CD16A and perforin expression, % CD2pos of CD16Apos NK cells and their CD2 expression.** The PBMCs were labeled with a panel of antibodies to identify the CD16Apos NK cells with CD2. Intracellular perforin was unimodal for NK cells identified as CD3negative perforin-positive lymphocytes. Protein expression was measured by the median fluorescence intensity (MFIs) of bound antibodies. Flow cytometric gating is depicted in Appendix A
Figure A4.

**Figure 4 antibodies-09-00054-f004:**
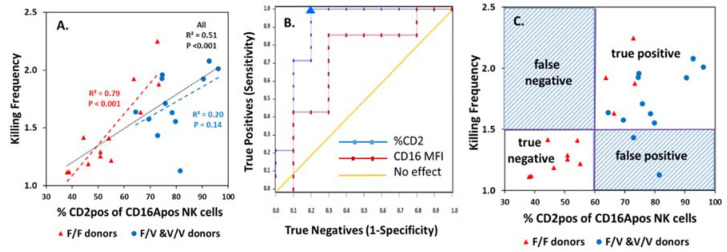
**Prediction of KFs by the %CD2pos of CD16Apos NK cells.** (**A**) **Correlation of %CD2pos of CD16Apos NK cells vs. KF.** Correlations for the CD16A genotypes are illustrated separately (red and blue) and combined (black dotted line). (**B**) **ROC analyses.** The sensitivities and specificities of a set of 24 samples for %CD2pos values and for CD16A MFIs (median fluorescent indices). The AUC was 0.89 for the %CD2pos (*p* < 0.001). The blue triangle indicates the donor with 63.7% CD2pos cells and represents the lowest %CD2pos that supported 100% sensitivity. (**C**) **Quadrants of a test using 60% CD2pos cells to predict a high KF.** The positive predictive value was 0.88.

**Table 1 antibodies-09-00054-t001:** Characteristics of the Donors.

Characteristic	All	Female	Male	CD16A AA158	
				F/F	F/V & V/V
Number of donors	24	16	8	12	12
Percentage	100%	67	33	50%	50%
Range, age in years	21 to 85	21 to 85	22 to 69	NA	NA
Mean age +/− sd	45 +/− 17	48 +/− 16	38 +/− 18	NA	NA
Median age	42	47	33	NA	NA

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
