# Peer review of "Natural Killer (NK) Cell Expression of CD2 as a Predictor of Serial Antibody-Dependent Cell-Mediated Cytotoxicity (ADCC)"

_2073-4468, 2020, doi:10.3390/antib9040054_

Round 1
Reviewer 1 Report
I appreciate your efforts to address all the reviewers comments and the subsequent improvement in the manuscript.
Author Response
Reviewer 1
Comments: I appreciate your efforts to address all the reviewers comments and the subsequent improvement in the manuscript.
Response: Thank you for your attention to the manuscript and its resubmissions. It is now clearer and suitable for a wide audience of scientists and clinicians.
Reviewer 2 Report
Scientifically, the manuscript is fine. However, it is disappointing that poor sentence structuring can be found at several places in the manuscript. The quality of the manuscript can be greatly improved by addressing this.
Example 1 - In lines 314-315, the authors mention 'The ‘effector’ (E) cytotoxic lymphocytes were freshly isolated, unstimulated blood NK cells with CD16A receptors for antibodies.' This sentence can written as follows: 'The ‘effector’ (E) cytotoxic lymphocytes were freshly isolated, unstimulated peripheral blood NK cells that express CD16A'
Example 2 - In lines 340-342, the authors say 'The KFs can be directly determined for each E:T in the ADCC assays, by dividing the percentage of target cells killed by the % dead expected if each cell in the cytotoxic cell population killed only one target cell.'. Two points: i) the authors should use 'percentage' or '%' but not both; ii) the second part of the statement is confusing and can be written as follows: '% dead expected, which assumes that each CD16Apos NK cell kills only one target cell.'
Example 3 - In line 368, the authors write 'The KFs at an E:T of 1:4 CD16Apos NK: Daudi'. The authors should either mention E:T or CD16Apos NK: Daudi, but not both.
Author Response
Reviewer 2
Thank you for your keen attention to the manuscript and its resubmissions. The wording that you flagged has been corrected. The manuscript is without problems that can be detected by Microsoft Word spell and grammar software or that were detectable by a colleague who is a journal editor. As a result of your efforts, the paper is now clearer. It is more readable by a wider audience of scientists and clinicians.
Comments: Scientifically, the manuscript is fine. However, it is disappointing that poor sentence structuring can be found at several places in the manuscript. The quality of the manuscript can be greatly improved by addressing this.
Example 1 - In lines 314-315, the authors mention 'The ‘effector’ (E) cytotoxic lymphocytes were freshly isolated, unstimulated blood NK cells with CD16A receptors for antibodies.' This sentence can written as follows: 'The ‘effector’ (E) cytotoxic lymphocytes were freshly isolated, unstimulated peripheral blood NK cells that express CD16A'
Response: Lines 314-315 are revised as requested and now read, “The ‘effector’ (E) cytotoxic lymphocytes were freshly isolated, unstimulated blood NK cells that express CD16A receptors for antibodies.’
Comments: Example 2 - In lines 340-342, the authors say 'The KFs can be directly determined for each E:T in the ADCC assays, by dividing the percentage of target cells killed by the % dead expected if each cell in the cytotoxic cell population killed only one target cell.'. Two points: i) the authors should use
'percentage' or '%' but not both; ii) the second part of the statement is confusing and can be written as follows: '% dead expected, which assumes that each CD16Apos NK cell kills only one target cell.'
Response: Lines 340-342 now read, “The KFs can be directly determined for each E:T in the ADCC assays, by dividing the % target cells killed by the % dead cells that would be expected under the assumption that each cell in the cytotoxic cell population kills only one target cell”.
Comments: Example 3 - In line 368, the authors write 'The KFs at an E:T of 1:4 CD16Apos NK: Daudi'. The authors should either mention E:T or CD16Apos NK: Daudi, but not both.
Response: Line 368 now reads, “The KFs at a 1:4 ratio of CD16Apos NK to Daudi cells were selected for inter-donor comparisons.” I checked and the manuscript lacks the noted redundancy in its other sections.
This manuscript is a resubmission of an earlier submission. The following is a list of the peer review reports and author responses from that submission.
Round 1
Reviewer 1 Report
Introduction requires some significant explanation of the concepts, the reviews highlighted could be used to incorporate more information for the reader to understand the concepts presented in the manuscript. Not just NKs or macrophages express CD16 (neutrophils?), and many other immune cells can perform ADCC - T cells for example. I thought CD16 was only found on a subset of circulating NK cells? explain? I'm unclear on the rationale and previous work that suggests ADCC is a possible cause of lack of response to antibody therapies in cancer - explain?
The experiments in figure 1 use whole PBMC, and yet figure 1 and 2 discuss NK responses - how do you know it's not just NK cells killing via CD16A? gamma delta T cells, CD8 T cells, NKT cells express CD16A - have you looked at the frequency of CD3- CD56bright/dim CD16A+ NK cells in each donors PBMC?
The figure should describe the experiments performed not the result, this should be in the main body of the text.
Figure 2: should you not correlate the frequency of CD16A+ cells in PBMC not just MFI. I don't know how you performed any of these experiments, please expand your methods.
Figure 3: I don't know how you did this, CD56+ NK cells? expression of CD16, perf, CD2?
Figure 4: The ROC analysis suggests both CD16 and CD2 are good predictors of True positives.
Reviewer 2 Report
Tang et al have carried out an elegant study to determine the ability of CD16 genotype, perforin levels and %CD2 positive effector cells to predict the killing frequency (KF) of NK cells in the presence of excess numbers of antibody-opsonized target cells. The authors demonstrate that only %CD2 positive effector cells correlate with KF. This is a very important and timely study. The manuscript can be further improved by addressing the following concerns/questions.
General comments:
The authors should comment on two potential limitations of the study: i) target cells were opsonized, washed and exposed to PBMCs. Even though the employed antibody is not internalizing, it is possible that the antibody has a off-rate of less than 4 hours, and the opsonized cells may lose the antibody overtime (non-physiological), limiting ADCC or KF; ii) Figure A4 shows that there are also significant number of monocytes in the PBMCs - these cells can either perform phagocytosis (leading to cancer cell death and Cr release) and/or trogocytosis (leading to reduction in ADCC).
Specific comments:
1) Several grammatical errors and suboptimal sentence structuring can be found in the manuscript. For example, 'to' in line 57 (at the end) should be removed. Such issues should be corrected.
2) Ref. 2, 3 are not sufficient for the statement made in the lines 62-63.
3) 'Intra-donor' is mentioned in several places, but I think it should be 'inter-donor' in most places.
4) Based on the data in reference 7, NKp46, NKG2D and DNAM-1 (in addition to CD2) synergize with CD16 in the context of crosslinking-induced activation. The authors should provide a reason for selecting CD2 but not the other receptors for this particular study.
5) Line 108 should be moved to the last paragraph of Introduction.
6) The sentence in lines 115-116 should include a reference (for perforin levels among individuals) and should be reformatted to clearly indicate that the 'three proteins' mentioned here include, CD16, CD2 and perforin.
7) In Table 1, the mean killing frequency should be excluded and included in the Results.
8) In line 260, 'Methodology' can be excluded.
9) It is not clear why E:T was used in few figures, while T:E was employed in the others. Technically, this is not an issue, but for consistency sake, the authors should use one format.
10) The authors should include 'Figure A2. KF increases as more targets are available' for all the donors.
11) Lines 314-317 are not clear (the authors should expand/rephrase it) - it is not clear why E:T of 1:4 was selected. To increase the confidence in the data, at least two different E:T should be used throughout.
12) In Figure 2B, what E:T was used? Was it 1:4?
13) In Figure 2C, did the authors employ an isotype control? It is important to understand whether the difference in MFI is due to the difference in the expression of CD16A or due to the difference in the binding of Fc region of staining antibody to CD16A.Also, did the donor with V/V genotype have the highest expression of CD16A?
14) In lines 387-390, it is not clear what the authors are aiming to convey. It is clear that %CD2+ can predict KF, but this test does not show whether immunotherapy will have an effect on the tumor cells. Therefore, all the patients will be suitable for receiving the therapy. It is possible that %CD2+ may help identify patients who will benefit more.
15) In line 394, the authors should describe what is the 'best cutoff CD16A MFI'.
16) The methodology used for plotting Figure 4B should be provided.
17) Lines 419-420 should be rephrased to improve the clarity.
18) In line 434, should it be '25%' instead of 50%?
19) Lines 442-446 should be followed by a comment to indicate that this could also be due to the use of afucosylated and engineered antibody.
20) The sentence in lines 470-472 is misleading - Several antibodies including Trastuzumab and Cetuximab can function through mechanisms independent of ADCC. Further, processes like trogocytosis by monocytes or macrophages can reduce the ADCC. In summary, the proposed test may reveal 'which patients may respond better' rather 'which patients will benefit'.
Reviewer 3 Report
Strengths of the manuscript:
- The topic of research is highly relevant and of significant interest for the research community. The identification of predictive markers for NK ADCC could lead to a better understanding of the molecular processes that are important for NK cell functionality in general and would theoretically enable a more targeted use of antibody-based therapies based of patient’s NK ADCC responsiveness.
Weaknesses of the manuscript:
- Language and style of the manuscript need major revisions. While the text is overly explanatory in some areas (e.g. flow cytometry marker selection, number of blood samples delivered per shipment,…), it is still quite difficult to understand and lacks essential information in other areas. E.g. the calculation of killing frequencies is difficult to understand from the methods section as well as the figure legend in Fig. 1, which also contains non legible labels. Other examples are Table 1 which does not state for what mean and median were calculated, as well as the unexplained differences in the calculation of percent specific 51Cr-release between the methods section and Fig. A1. Further, the use of exclamation marks in scientific manuscripts is inappropriate and the amounts of typos needs to be strongly reduced.
- The experimental design is not optimal and would profit from better explanations. E.g. it is claimed that the utilized ADCC technique is advantageous since it does not loose NK cells by working with whole PBMC populations instead and it is suggested that thus potentially more of the 6000+ phenotypes of NK cells are represented in the experiment. While it is great to preserve as much NK cell diversity as possible, untouched NK cell isolation methods are very efficient and the work with pure NK effector cells excludes potential influences of other immune cells as well as the rather complicated and unusual calculations that are made to adjust for varying NK cell numbers in the experiments. However, this relevant limitation is not discussed and more NK cells (and thus potentially more phenotypes of those cells) could be used per well using isolated NK cells. Another problematic statement is the claim that serial killing is observed. The utilized method can by set-up not measure serial events since it is a single time point that is measured. Just because more target cells than effector cells are killed does not mean serial killing occurred since this could also be the result of mere bystander cell killing especially when cell density is increased.
- The presentation of data is not focused and partially neither relevant to the manuscript nor helpful. E.g. it would be helpful to include a graphic explaining the ADCC method in Figure 1 but since age and gender were not relevant for ADCC the data doesn’t need to be shown. Another example: Figure 2 does not depict relevant information as there were no significant differences between the measured parameters detected beyond the level of CD16A expression, which is already known and even mentioned in the introduction.
- Use of statistical methods and interpretation of data. The use of statistical methods needs major revisions and claims need to be supported by and fit the data. E.g. regression curves should only be plotted if the data is sufficient to do so – for the small number of males in Fig. 1B there is not sufficient data and the results themselves state that this is not a fitting model to explain the data. For Fig. 4A on the other hand, the data shows that actually the %CD2pos cells just correlate with killing frequency of F/F donors, which suggests that %CD2 pos cells is actually only a potential predictor for ADCC for patients with this genotype. However, this is not discussed and instead a more universal claim is made. Another example: It is claimed that 4.4-fold more antibody was needed to support ADCC by FF donors even though the T-test results show no significant differences between the compared genotypes regarding their EC50. In paragraph 3.3. claims are also made about the use of the optimal antibody (assumingly meaning antibody concentration but what is this and where is the data supporting this claim) and that thus intra-donor differences were measured in conditions that over-rode CD16A effects that might occur at low concentrations (this is another assumption lacking data). In particularly the discussion needs to be revised to remove unsupported claims and reduce the significance that is associated with potentially relevant but very small amounts of data.
Overall recommendation:
In the current state of the manuscript, it is difficult to conceive whether or not the experimental findings will be convincing after mayor revisions.. While in my personal opinion the data is not sufficient for a full-lengths manuscript, I believe that after editing the information it might be sufficient for a relevant short report or letter. For this purpose, I suggest to focus solely on the findings that more CD2% cells are present in donors with high ADCC activity while CD16A and Perforin expression apparently are not relevant. I wish the authors the best of luck with their further work on this very relevant area of study.
Round 2
Reviewer 1 Report
Thanks for the detailed response. I would like to see further editing of the whole document. I am still concerned about the interpretations from a whole PBMC assay, however more robust justification of the experiments with reference to Sung et al will alleviate this somewhat. I still suggest that if possible data from sorted NK cells for a critical experiment be performed but without this removing overinterpretations of serial ADCC from these assay will suffice.
Reviewer 2 Report
General comments were not addressed satisfactorily: the authors use PBMC:Target ratio to compare 0h and 4h data. However, most of the data in the manuscript was derived using NK:Target ratio.
Grammatical errors/inappropriate sentence structuring still exists throughout the manuscript. Also, it is difficult to understand what the authors are trying to say in their response because of the grammar/sentence structure - for example, response to general comments and comment# 4 and 12.
The authors should improve the manuscript and most of the responses in the context of the above before I can evaluate the revised version.
Reviewer 3 Report
Dear authors,
thank you for providing comprehensive responses to my suggestions including literature references that were very helpful in understanding the authors aims with different aspects of manuscript. While the edits have improved the introduction to become better understandable, insufficient effort was made to improve experimental design and data as well as the figures. Furthermore, the text still needs more editing to become better understandable and typos are still present. Thus, I sadly cannot recomment to accept this manuscript at the present time.